# Insights into Layered Oxide Cathodes for Rechargeable Batteries

**DOI:** 10.3390/molecules26113173

**Published:** 2021-05-26

**Authors:** Julia H. Yang, Haegyeom Kim, Gerbrand Ceder

**Affiliations:** 1Department of Materials Science and Engineering, UC Berkeley, Berkeley, CA 94720, USA; juliayang@berkeley.edu; 2Materials Sciences Division, Lawrence Berkeley National Laboratory, Berkeley, CA 94720, USA; haegyumkim@lbl.gov

**Keywords:** layered oxide cathodes, alkali–alkali interactions, electronic structure, Li diffusion

## Abstract

Layered intercalation compounds are the dominant cathode materials for rechargeable Li-ion batteries. In this article we summarize in a pedagogical way our work in understanding how the structure’s topology, electronic structure, and chemistry interact to determine its electrochemical performance. We discuss how alkali–alkali interactions within the Li layer influence the voltage profile, the role of the transition metal electronic structure in dictating O3-structural stability, and the mechanism for alkali diffusion. We then briefly delve into emerging, next-generation Li-ion cathodes that move beyond layered intercalation hosts by discussing disordered rocksalt Li-excess structures, a class of materials which may be essential in circumventing impending resource limitations in our era of clean energy technology.

## 1. Introduction to the O3 Structure

Rechargeable Li-ion batteries have enabled a wireless revolution and are currently the dominant technology used to power electric vehicles and provide resilience to a grid powered by renewables. Research in the 1970s to create superconductors by modifying the carrier density of chalcogenides through intercalation [1] transitioned into energy storage when Whittingham demonstrated in 1976 a rechargeable battery using the layered TiS_2_ cathode and Li metal anode [2]. Soon thereafter, Mizushima, Jones, Wiseman, and Goodenough demonstrated that a much higher voltage could be achieved by reversible Li de-intercalation from layered LiCoO_2_ [3], energizing generations of rechargeable battery research. In this short review we revisit our work in understanding a few basic relationships between the structure, electronic structure, and properties of layered cathode materials. 

A layered rocksalt cathode oxide adopts the general formula A_x_MO_2_ (A: alkali cation, M: metal cation, O: oxygen anion). The O anions form a face-centered cubic (FCC) framework with octahedral and tetrahedral sites. These two environments are face sharing and form a topologically connected network. When fully alkaliated such that x ~ 1, the compound consists of AO_2_ and MO_2_ edge-sharing octahedra. The layered structure, illustrated in Figure 1, is aptly named because AO_2_/MO_2_ octahedra form alternating (111) planes of the FCC oxygen lattice when fully lithiated. The A and M cations alternate in the abc repeat unit of the oxygen framework to form a−b_c−a_b−c_*,* stacking where the minus sign “−” indicates the location of M and the underscore “_” gives the position of the A ions. Because the oxygen stacking has a repeat unit of three and the metal layering repeats every two layers, periodicity is achieved after six oxygen layers. Under the structural classification by Delmas et al. for layered cathode oxides [4], the layered rocksalt cathode structure is commonly referred to as O3: O for the octahedral alkali ion environment (not to be confused with O for oxygen) and 3 for the number of MO_2_ slabs in a repeat unit. The O3 structure is equivalent to the structure of α-NaFeO_2_ and the cation ordering is also known in metallic alloys as L1_1_ (CuPt prototype) [5].

It is now well understood that the ordering of AO_2_ and MO_2_ in alternating layers is not the most favored cation ordering from an electrostatic perspective. Instead, the layered structure finds its stability in the size difference of A and M [6,7] as it allows A-O and M-O bond distances to relax independently of each other. This independent A-O and M-O bond accommodation explains how a larger A cation, such as Na^+^, can form the layered structure with a wide range of M radii [8], whereas a smaller A, such as Li^+^, only forms stable O3 compounds with a limited range of smaller M radii, namely Co^3+^ [9], V^3+^ [10], Ni^3+^ [11], and Cr^3+^ [10].

## 2. Evolution from LiCoO_2_ to NMC

Today, O3 cathodes have evolved in several directions from LiCoO_2_ [12] for use cases beyond portable electronics. Anticipating potential cost and resource problems with Co [13], research in the 1980s and 1990s mostly focused on substitutions of Co by Ni [14]. However, consideration of the low cost of Mn and the high stability of the Mn^4+^ charge state led the community towards layered LiMnO_2_. Even though this structure is not the thermodynamically stable state of LiMnO_2_ [15], Delmas [16] and Bruce [17] were able to synthesize it by ion exchange from the stable NaMnO_2_. Unfortunately, the high mobility of Mn^3+^ [18] leads to a rapid transformation of the layered structure into the spinel structure upon cycling [19] because of its pronounced energetic preference at the Li_0.5_MnO_2_ composition [20]. Attempts to stabilize layered LiMnO_2_ with Al [21] or Cr [22,23] substitution were only partially successful and led to the formation of a phase intermediate between layered and spinel [24]. Then, in 2001, several key papers were published that would pave the way for the highly successful Ni-Mn-Co (NMC) cathode series: Ohzuku showed very high capacity and cyclability in Li(Ni_1/3_Mn_1/3_Co_1/3_)O_2_ [25], known as NMC-111, and in Li(Ni_1/2_Mn_1/2_)O_2_ [26]; Lu and Dahn published their work on the Li(Ni_x_Co_1−2x_Mn_x_)O_2_ [27] and its Co-free Li-excess version Li(Ni_x_Li_1/3−2/x_Mn_2/3−x/3_)O_2_ [28]. In these compounds Ni is valence +2 and Mn is +4 [29], thereby stabilizing the layered material against Mn migration and providing double redox from Ni^2+^/Ni^4+^. At this point the NMC cathode series was born. Since then, Ni-rich NMC cathodes have become of great interest to both academia and industry because they deliver a capacity approaching 200 mAh/g and demonstrate high energy density, good rate capability, and moderate cost [30,31,32].

In this short article, we summarize some general and fundamental understanding we have gained in layered oxide cathodes, without delving into issues with very specific compositions. We focus on the roles of the alkali–alkali interaction, electronic structure, and alkali diffusion, and illustrate how these fundamental features conspire to control the electrochemical behavior of O3-structured layered oxides.

## 3. Alkali–Alkali Interactions, Alkali/Vacancy Ordering, and Voltage Slope

The voltage of a cathode compound is set by the chemical potential of its alkali ions [33] which itself is the derivative of the free energy with respect to alkali concentration. This thermodynamic connection between voltage and free energy creates a direct relation between the voltage profile, the alkali–alkali interactions, and phase transformations as functions of alkali content. While Na_x_MO_2_ compounds show many changes in the stacking of the oxygen host layers when the Na content is changed, phase transitions in Li_x_MO_2_ materials are mostly driven by the Li-vacancy configurational free energy, resulting from Li^+^-Li^+^ interactions in the layer [20,34]. In layered compounds with a single transition metal, such as Li_x_CoO_2_ and Li_x_NiO_2_, such phase transitions are easily observed as voltage plateaus and steps in the electrochemical charge–discharge profiles as shown in Figure 2a. For a first-order phase transformation, for example from Phase I to Phase II, the Gibbs phase rule dictates that the Li chemical potential should be constant, hence the voltage remains constant while one phase transforms into the other. Phases in which the alkali ions are well-ordered usually display a rapid voltage change as the alkali content is changed, reflecting the high energy cost of trying to create off-stoichiometry in ordered phases. This is in contrast to solid solutions which have smoother voltage profiles as a function of alkali concentration. For example, both theory [35] and experiments [36,37] indicate that in Li_x_CoO_2_ a monoclinic phase appears with lithium and vacancies ordered in rows for x ≈ 0.5 [36]. In Li_x_NiO_2_, Li-vacancy ordering is responsible for stable phases at x ~ 0.8, ~0.5, and ~0.25–0.3 [38,39,40]. When many transition metals are mixed, as in NMC cathodes, the Li^+^-Li^+^ interaction remains present, but Li-vacancy ordering is suppressed by the electrostatic and elastic perturbations on the Li site caused by the distribution of the Ni, Mn and Co in the transition metal layers.

The Li^+^-Li^+^ interaction is mostly electrostatic but is highly screened by the charge density on the oxygen ions, leading to a rather small effective interaction in layered Li_x_MO_2_ compounds and small voltage slope. This is a critical feature of Li_x_MO_2_ compounds that gives them high capacity in a relatively narrow voltage window compared to other alkali compounds, as explained below. The effective interaction between intercalating ions increases significantly when larger alkali ions (e.g., Na^+^ and K^+^) are used in the layered structure [41,42,43]. These larger alkali ions increase the oxygen slab distance, reducing the oxygen charge density available for screening within the alkali layer [20,42,43]. The larger effective repulsion between the Na^+^ or K^+^ ions affects the phase transition and electrochemistry in a very significant way as shown in Figure 2b. For example, Na*_x_*CoO_2_ has stronger Na-vacancy ordering and thus more pronounced voltage steps compared to Li*_x_*CoO_2_. This phenomenon [44,45] becomes even more significant in K*_x_*CoO_2_ [46,47]. The effect of the intercalant’s size on the phase transitions and voltage steps is not just limited to Co-containing compounds but is also generally applicable to other transition metal systems as described in a recent review [43].

In practice, the larger effective interaction between alkali ions in the layered transition metal oxides is detrimental for their electrochemical performance. First, more phase transitions are likely to induce more mechanical stress in the cathode structure during charging and discharging, causing possible fracture of electrode particles. Second, a simple argument shows that the average voltage slope is proportional to the effective interaction: V(x) is equal to −μ_Li_(x), and since μ_Li_(x) = $\frac{\partial G}{\partial x}$, $\frac{\partial V}{\partial x}$ = −$\frac{\partial^{2}G}{\partial x^{2}}$. In a simple regular solution model for mixing, this second derivative of the free energy is proportional to the effective interaction [41,43,49]. Hence, when the effective interaction is large, as in layered K_x_MO_2_ compounds, the voltage curve has a high slope, limiting the achievable capacity between fixed voltage limits. This analysis shows that the advantage of lithium systems in providing large capacity within reasonable voltage limits is in part due to the highly effective screening of the Li^+^-Li^+^ interaction by oxygen. For Na, and in particular for K-ion based intercalation energy storage, it may be more advantageous to search among poly-anion compounds for good cathodes [43].

## 4. Electronic Structure of LiCoO_2_

The electronic structure of layered LiMO_2_ oxides is well understood. Due to the large energy difference in electronic levels between Li and the transition metal (TM), their electronic states do not mix and the behavior of the compound is controlled by the (MO_2_) complex within which the transition metal and oxygen hybridize. In the R3-m symmetry of the layered structure, the environment of the TM is pseudo-octahedral in that all TM-O bond lengths are of equal length, but O-TM-O angles have small deviations from those in a perfect octahedron. The pseudo-octahedral symmetry splits the otherwise degenerate TM-*d* orbitals in three (lower energy) t_2g_ and two (higher energy) e_g_ orbitals yielding an energy separation called the octahedral ligand-field splitting, abbreviated as $\Delta_{0}$. A more complete schematic of an orbital diagram is given in Figure 3a. The t_2g_ orbitals are shown as “non-bonding” in this schematic though in reality some π-hybridization takes place between them and the oxygen *p*-orbitals [50]. In the most basic picture in which one considers the overlap of the TM-*d* orbitals with its ligand *p*-states, the t_2g_ orbitals are formed from the *d*_xy_-type *d*-orbitals which point away from ligands. In contrast, the *d*_z2_ and *d*_x2–y2_ orbitals of the TM point toward the ligand creating *σ*-overlap. The e_g_* orbitals are the anti-bonding component of this hybridization and are dominated by TM states, whereas the bonding components, e_g_^b^, sit deep in the oxygen-dominated part of the band structure. Hybridization of the oxygen 2*p* and the metal 4*d* and 4*s* make up the remaining part of the band structure. Because the e_g_* orbitals result from σ-overlap between TM *d* states and oxygen *p*-states their energy is most sensitive to the TM-O bond length. Inducing, for example, a Jahn–Teller distortion moves these levels considerably. One can recover the characteristics of this molecular orbital diagram in a more realistic band structure and density of states computed with Density Functional Theory using the meta-GGA SCAN density functional approximation [51] as shown in Figure 3b for LiCoO_2_. The elemental contributions are indicated by the color of the bands with green being oxygen and red the Co 3*d* states. The two e_g_*-like bands above the Fermi level (solid line) have mixed Co and O contribution while the three t_2g_-like bands below the Fermi level have more pure metal contribution. These three bands have a small bandwidth due to their non-bonding nature. The lowest six bands shown are the bands dominated by the oxygen states in green.

## 5. Electronic Structure Trends in Layered Li_x_MO_2_ Oxides

Because the ligand-induced splitting between transition metal *d* orbitals is fairly small, filling of states usually follows Hund’s rule for creating high spin ions. Figure 4 illustrates this filling for 3*d* octahedral transition metal ions. Examples of this are Fe^3+^ (*d*^5^) and Mn^4+^ (*d*^3^). The high spin band filling implies that Mn^3+^ (*d*^4^) and Fe^4+^ (*d*^4^) with a single occupied e_g_* state are Jahn–Teller active ions [50,53]. The later transition metals form exceptions to the high-spin rule in that Co^3+^ and Ni^4+^ are low-spin *d*^6^ with all electrons occupying t_2g_ states. We discuss below that this is a key reason for their predominance as redox-active materials in layered oxides. The lack of any filled antibonding states in Co^3+^ makes this cation also one of the smallest 3*d* TM ions, which is reflected in the very high crystal density of LiCoO_2_ of 5.051 g/cm^3^ [54] and the associated high energy density. This electronic structure-induced high density makes LiCoO_2_ still the preferred cathode material for portable electronics where battery volume comes at a high premium.

Upon Li removal the hybridization of the orbitals changes. As an electron is removed from TM states the remaining TM *d* electrons experience less intra-atomic Coulombic repulsion and their states move down in energy, bringing them closer to the oxygen *p* states, resulting in increased hybridization. There are two notable consequences from this rehybridization [51,55]: (1) As hybridization transfers (filled) oxygen states onto the metal it increases the electron density on the metal site. Somewhat counterintuitively, almost no electron density change occurs on the metal after delithiation, even though it is formally oxidized, something that had been recognized earlier outside of the battery field in various Mn-oxides. This rehybridization by the anion explains why the anion has an almost larger influence on the voltage than the choice of 3*d* TM ion in layered compounds [52]. Effectively, the flexible hybridization between the TM and the O ligand creates a charge density buffer on the TM. (2) Covalency increases upon charging, leading in some cases to the fully charged material taking on the O1 (octahedral alkali environment with a repeat unit of 1 [4]) structure which is typically found in more covalent materials such as CdI_2_. The increased covalency also sharply contracts the Li slab spacing (distance between the oxygen layers around the Li-layer) and *c*-lattice parameter when most of the alkali is removed [56].

## 6. Implications of Electronic Structure on Layered Stability

The orbital filling of the transition metals also plays a critical role in the stability of the layered structure upon Li removal. Ions with filled t_2g_ levels are most stable in the octahedral environment and resist any migration into the Li layer [57]. Because the oxygen arrangement is topologically equivalent to an FCC lattice, octahedral cation sites edge-share with each other and face-share with a tetrahedral site. Since ion migration through a shared edge comes with a very high energy barrier, cation diffusion between octahedral sites requires passage through an intermediate tetrahedral site. For ions with filled t_2g_ states, this passage through the tetrahedral site and the shared anion face raises the energy substantially as the octahedral ligand field stabilization is lost, making these ions all but immobile. In contrast, ions with *d*^5^-high-spin (e.g., Mn^2+^ or Fe^3+^) and *d*^0^ filling (e.g., Ti^4+^, V^5+^) tend to be much more ambiguous about their preferred anion coordination, and as a result, tend to migrate more easily [57,58,59]. The most stable octahedral cations are therefore low-spin-Co^3+^ (*d*^6^) and low-spin-Ni^4+^ (*d*^6^). In addition, high-spin-Mn^4+^ (*d*^3^) also possesses very high octahedral stability as it adds Hund’s rule coupling to the ligand field stabilization. These insights allow us to rationalize the prominence of the NMC class of layered oxides as cathodes in the Li-ion industry: Only Ni and Co have very high resistance against migration into the Li layer in the charged and discharged states. Mn^4+^ acts similarly, but cannot be used as a redox active element as its reduction or oxidation leads to an ion that is prone to migration [57,60,61]. Within the 3*d*-TM series there are unfortunately no other ions which can match the octahedral stability of the NMC chemistry, and layered oxides based on other *3d* TM are unlikely to be practical. Hence, it is the basic electronic structure of the 3*d* transition metal ions which is the direct cause of the serious resource problem the Li-ion industry faces if it wants to scale to multiple TWh annual production with layered oxides [13,62].

## 7. Diffusion Mechanism

In understanding alkali transport in layered compounds, and more generally in closed-packed oxides, it is important to assess how structure and chemistry influence performance, and ultimately, how one can design novel dense cathodes that are not layered. In this section, we focus on Li diffusion. Even though it is likely that Na and K migrate through a similar pathway, much less work has been done to validate the transport mechanism of these larger alkalis. The octahedral-tetrahedral-octahedral topology introduced earlier determines the diffusion mechanism in layered materials. Lithium migrates along the minimum energy path between two stable sites via an activated state, with the activation barrier defined as the difference between the maximum energy point along the path and the initial equilibrium position of the ion [63]. In layered Li_x_CoO_2_ [35], migration between neighboring octahedral sites can in principle occur through the shared octahedral edge formed by an oxygen–oxygen dumbbell (a mechanism referred to as an octahedral dumbbell hop (ODH)), or via the tetrahedral site that faces-shares with the initial and final octahedron (a tetrahedral site hop (TSH)). Figure 5 illustrates the ODH and TSH mechanisms in Li_x_CoO_2_.

A TSH requires the presence of a divacancy in the Li layer, which in Figure 5, implies that both s_f_ and NN1 need to be vacant. Van der Ven et al. used ab initio calculations to show that the ODH mechanism has a considerably higher activation barrier (~800 meV) than a TSH mechanism (230–600 meV) [63], establishing that Li diffusion in Li_x_CoO_2_ occurs predominantly by way of TSH for all practical lithiation levels. Even though the Li^+^ in the activated state in the tetrahedron faces repulsion from a face-sharing Co^3+/4+^ ion, the large Li slab spacing keeps the distance between Co and Li reasonable (Appendix A).

The barrier for the TSH mechanism can vary from 230 to 600 meV due to local environment changes during lithiation. At x ~ 0.5, Li diffusion dips due to the ordering reaction [35] in agreement with experiment [36,64,65]. When a larger amount of Li (0.5 > x in Li_x_CoO_2_) is removed from the compound, a large decrease in the *c*-lattice parameter is observed experimentally and from first-principles [66,67,68]. Such lattice contraction increases the activation barrier significantly because it creates a smaller tetrahedron height and shorter distance between the activated Li^+^ and the Co^3+/4+^ ion. The larger positive charge on Co when the compound is more oxidized also contributes to an increase of the energy in the activated state. As a result, the activation barrier increases by hundreds of meV when delithiation increases past 0.5 Li [69]. While the precise behavior of the c-lattice parameter and slab spacing in NMC materials depends on the specific chemistry, the overall behavior is similar to LiCoO_2_.

## 8. Beyond Layered Materials: DRX

It is now understood that the Li migration mechanism in layered oxides is a specific case of a more general framework for understanding ion transport in FCC close-packed oxides. Recent work [70,71] categorized the different environments that can occur around the tetrahedral activated state in close-packed oxides by the number of face-sharing transition metals it has. So-called *n*TM channels have *n* transition metals face-sharing, with the other face-sharing octahedral sites either occupied by Li or vacancies. Because minimally two Li (or vacant sites) are required to create a migration path, 4TM and 3TM channels do not participate in diffusion. Structures with only 2TM channels exist but display very poor Li mobility due to the large electrostatic repulsion Li^+^ sees in the activated state from the two TM ions with which it face-shares. This theory explains why ordered γ-LiFeO_2_, a compound with only 2TM channels, is not electrochemically active. Layered oxides contain 3TM and 1TM channels with Li diffusion occurring through the 1TM channels. However, the proximity of the TM to the Li^+^ in a 1TM channel creates a strong dependence of the migration barrier on the size of the tetrahedron, which in layered materials is determined by the slab spacing. As shown extensively by Kang et al. [72,73], even small contractions of the slab spacing, caused by TM mixing in the Li layer, reduce Li mobility in a very substantial way. The “safest” migration paths are 0-TM channels: In the presence of a divacancy, a migrating Li^+^ only electrostatically interacts with one other Li^+^ ion making the activation energy rather insensitive to dimensional changes. Recent work has shown how to create cation-disordered materials in which transport occurs through these 0-TM channels [74]: When cations are fully disordered over the octahedral sites of a structure with FCC anion packing, all possible configurations around the activated state occur with some statistical probability, and percolation of the 0-TM channels into a macroscopic diffusion path occurs when more than 9% Li excess is present (i.e., x > 0.09 in Li_1+x_M_1-x_O_2_). In reality, cation short-range order tends to reduce the amount of 0-TM channels from what would be in a random system [75] and a higher Li-excess content is needed or high-entropy ideas have to be applied to minimize short-range order [76]. Based on these insights, Li-excess rocksalt oxides with a disordered cation distribution have been shown to function as intercalation cathodes with high capacity [77]. Disordered Rocksalt Li-eXcess cathodes (DRX), also referred to as DRS (Disordered RockSalt) by some [78], do not require any specific ordering of the metal and Li cations, and can therefore be used with a broad range of redox active and non-redox active metals, alleviating the resource issue arising with NMC layered cathodes. Indeed, almost all 3*d* and several 4*d* TM have been used as redox couples, including V^4+/^V^5+^ and partial V^3+^/V^5+^ [79], Mn^2+^/Mn^4+^ [80], Mo^3+^/Mo^6+^ [81], and partial Fe^3+^/Fe^4+^ [82,83]. Redox-inactive *d*^0^ TM elements, such as Ti^4+^, V^5+^, Nb^5+^, and Mo^6+^ [77], play a particular role in DRX as they stabilize disorder [84], and their high valence compensates for excess Li content. Fluorination (anion substitution) is possible in DRX compounds which lowers the cation valence and extends cycle life by reducing oxygen redox [85]. Promising specific energies approaching 1000 Wh/kg (cathode only) have been achieved with Mn-Ti-based materials offering a possible low-cost, high-energy cathode solution for Li-ion.

## 9. Conclusions

The pioneering work of Professor Goodenough on LiCoO_2_ [3] has led to a rich and widely used class of layered cathodes thereby transforming Li-ion into the leading energy storage technology for electronics, vehicles, and the grid. In this review, we discussed the topology of the layered structure and explain how the structure (1) sets the voltage slope trends among various alkali ions, (2) is critically limited to certain transition metals due to their electronic structure, and (3) controls the alkali diffusion mechanism. A 20-year effort to understand the phase stability, transport, and electronic structure in these compounds can now be broadened towards new high-energy density cathode materials, ensuring the future of Li-ion as an important contribution to clean energy technology.

## Figures and Tables

**Figure 1 molecules-26-03173-f001:**
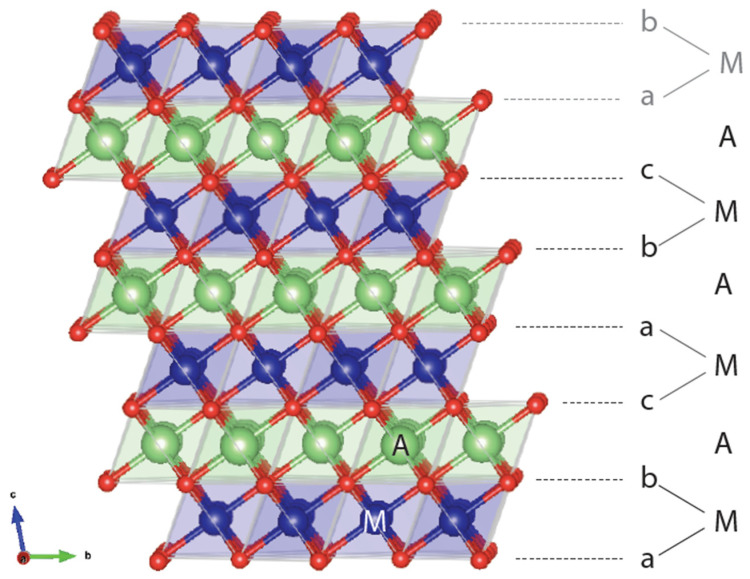
Representative O3 structure showing the abc stacking sequence of oxygen ions (red), thus creating various coordination environments for the alkali ion A (green) and metal ion M (blue). In an O3 repeat unit, M and A are coordinated below and above by oxygen layers. The beginning of another repeat unit with a-M-b stacking is in gray.

**Figure 2 molecules-26-03173-f002:**
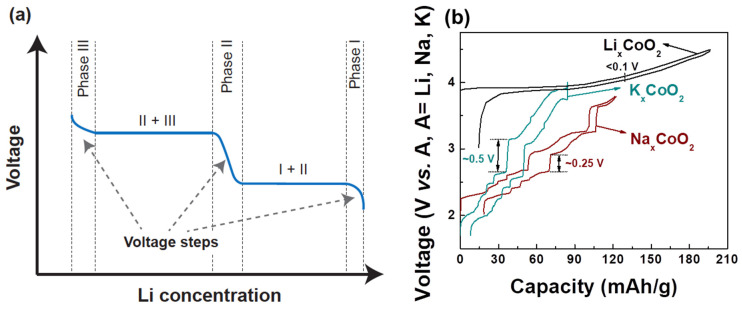
(**a**) Typical charge–discharge of intercalation-based cathode materials. A voltage step indicates new phase formation. (**b**) Charge–discharge comparison of O3-Li_x_CoO_2_, P2-Na_x_CoO_2_, and P2-K_x_CoO_2_ [44,47,48]. Voltage curves for P2-Na_x_CoO_2_, P2-K*_x_*CoO_2_, and O3-Li_x_CoO_2_ are reproduced with permissions from [44,47,48]. The voltage curve for P2-K_x_CoO_2_ is licensed under CC BY-NC 4.0 [47].

**Figure 3 molecules-26-03173-f003:**
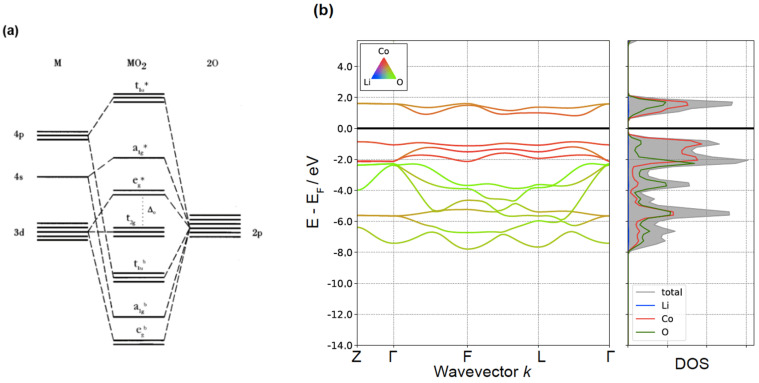
(**a**) The molecular orbital diagram for a (MO_2_)^−^ complex in octahedral environment. Reproduced with permission from [52]. (**b**) Calculated band structure and density of states with projections onto local orbitals for LiCoO_2_, showing elemental contributions from Li (blue), Co (red), and O (green).

**Figure 4 molecules-26-03173-f004:**
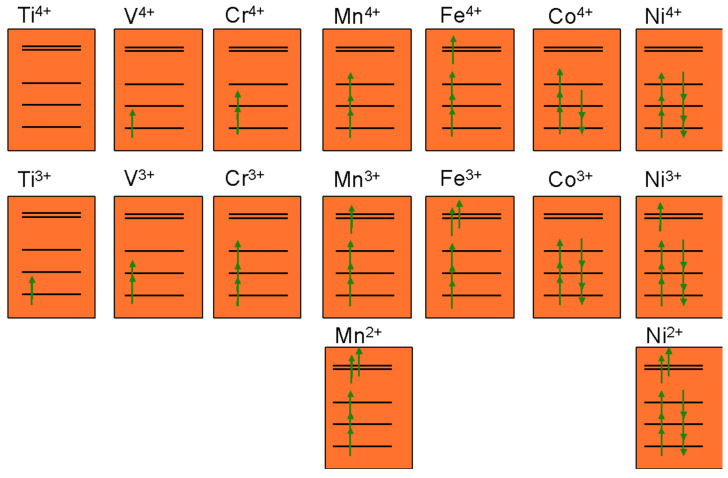
Expected transition metal band filling in the three t_2g_ and two e_g_* states for 3*d* transition metals in octahedral environments.

**Figure 5 molecules-26-03173-f005:**
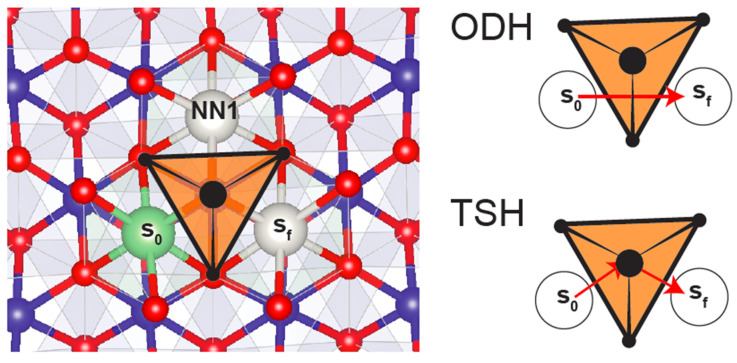
Li diffusion from site s_0_ to s_f_ in layered materials. View of Li_x_CoO_2_ down the *c*-axis, with the top CoO_2_ metal layer removed for clarity. The Li (green) layer and CoO_2_ metal layer (Co: blue, O: red) below are shown. Black circles further indicate the O which coordinate the tetrahedral site colored in orange, showing how three of the tetrahedron faces are face-sharing with s_0_, NN1, and s_f_. The last face of the tetrahedron face-shares with Co in the metal layer below. Li in site s_0_ can either diffuse from octahedral s_0_ to octahedral s_f_ via the edge-sharing connection, thus completing an octahedral dumbbell hop (ODH) illustrated by the single red arrow, or through the empty tetrahedral site via the tetrahedral site hop (TSH) mechanism illustrated by the two red arrows. Site NN1 (white) is vacant.

## Data Availability

The LiCoO_2_ structure used for calculating the band structure is freely available via the Materials Project database at https://materialsproject.org (accessed on 26 May 2021).

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
