# Peer review of "Insights into Layered Oxide Cathodes for Rechargeable Batteries"

_molecules, 2021, doi:10.3390/molecules26113173_

Round 1

Reviewer 1 Report

In this short review, the authors of the manuscript succinctly summarized their understanding of cathode materials for rechargeable Li-ion batteries. The work presented in the manuscript is of scientific interest and suitable for the readers of the journal. However, the authors may be advised to update their manuscript according the following comments.

  1. The abbreaviations for oxygen and octahedral are conflicting.
  2. Figures in Figure 3 are not labeled as (a) and (b).
  3. Figure 5 is not cited in the main text.

Reviewer 2 Report

Lithium-ion batteries have been applied extensively in many fields and the cathode materials are essential for the performances.

In this review paper, the authors summarized and described the layered lithium oxide compounds for the cathodes of rechargeable Li-ion batteries. The influence of morphology, chemistry and electronic structure on the electrochemical behavior of them are discussed in detail. The topic is of great significance and The paper is well written.

I suggest it can be published after some minor revision.

1. The abstract should be modified. References should be removed in the abstract.

2. The re-used pictures, for example Fig. 1, Fig. 4 and Fig. 5, must mention their sources.

3. The first presence of the abbreviations such as NMC should show readers the full names.

4. The outlook of the materials may be added finally.

Reviewer 3 Report

The authors summarized their understanding of how the structure’s topology, electronic structure, and chemistry interact to determine the electrochemical performance of layered intercalation compounds cathode materials in a pedagogical way. The following remarks and comments have to be taken into account for the acceptation of the article.

  1. The title is “Insights into layered lithium oxide cathodes for rechargeable batteries”, but it seems that your research is about AxMO2 ternary metal oxides and NMC cathodes. The use of “lithium oxide” is incorrect and not general, so it’s better to modify the title and reflect the generality.
  2. The abstract is too long, and it’s better to adjust the part of the research history to the main text. Moreover, you have to refine the abstract to highlight what research you have done.
  3. At present, the content of each part in the main text is not completely j You’d better re-divide the main text by setting some headlines and subtitles. The headline “3. Interaction between alkali ions and consequent alkali/vacancy ordering and voltage slope in the layered structure” is too long and should be refined.
  4. No explanation is given when some abbreviations, such as “NMC” and “TM”, appear for the first time, and please modify the manuscript carefully.
  5. The conclusion section does not summarize to the full text. Since you have studied how the structure’s topology, electronic structure, and chemistry interact to determine the electrochemical performance, which material is the best based on your present research?
  6. The writing and language need to be greatly improved.

Round 2

Reviewer 3 Report

The author revised all the questions, the paper can be published now.